# The Occurrence of Antimicrobial-Resistant *Salmonella*
*enterica* in Hatcheries and Dissemination in an Integrated Broiler Chicken Operation in Korea

**DOI:** 10.3390/ani11010154

**Published:** 2021-01-11

**Authors:** Ke Shang, Bai Wei, Se-Yeoun Cha, Jun-Feng Zhang, Jong-Yeol Park, Yea-Jin Lee, Hyung-Kwan Jang, Min Kang

**Affiliations:** 1Department of Veterinary Infectious Diseases and Avian Diseases, College of Veterinary Medicine and Center for Poultry Diseases Control, Jeonbuk National University, Iksan 54596, Korea; shangke0624@gmail.com (K.S.); weibai116@hotmail.com (B.W.); kshmnk@hanmail.net (S.-Y.C.); jfzhang018@gmail.com (J.-F.Z.); jyp410@naver.com (J.-Y.P.); lyj95923@naver.com (Y.-J.L.); 2Bio Disease Control (BIOD) Co., Ltd., Iksan 54596, Korea

**Keywords:** *Salmonella enterica*, antimicrobial resistance, hatchery, integrated broiler chicken operation, transmission

## Abstract

**Simple Summary:**

The present study investigated the emergence, antimicrobial susceptibility profiles, serotypes, and genotypes of *Salmonella*
*enterica* from hatcheries and their upstream breeder farms to determine the occurrence of antimicrobial-resistant (AMR) *Salmonella*
*enterica* contamination in hatcheries and its dissemination in an integrated broiler chicken operation. The hatcheries showed a high prevalence of *Salmonella* isolates with high antimicrobial resistance and no susceptible isolate. The AMR isolates from hatcheries originating from breeder farms could disseminate to the final retail market along the broiler chicken supply chain. Hatcheries play a more important role than breeder farms in the occurrence and spread of *Salmonella* in the broiler production chain. The emergence of AMR *Salmonella* in hatcheries may be due to the horizontal spread of resistant isolates rather than the first contamination of susceptible *Salmonella* and acquisition of resistance. Therefore, *Salmonella* control in hatcheries, particularly its horizontal transmission, is important.

**Abstract:**

Positive identification rates of *Salmonella enterica* in hatcheries and upstream breeder farms were 16.4% (36/220) and 3.0% (6/200), respectively. Among the *Salmonella* serovars identified in the hatcheries, *S*. *enterica* ser. Albany (17/36, 47.2%) was the most prevalent, followed by the serovars *S*. *enterica* ser. Montevideo (11/36, 30.6%) and *S*. *enterica* ser. Senftenberg (5/36, 13.9%), which were also predominant. Thirty-six isolates showed resistance to at least one antimicrobial tested, of which 52.8% (*n* = 19) were multidrug resistant (MDR). Thirty-three isolates (enrofloxacin, MIC ≥ 0.25) showed point mutations in the *gyr*A and *par*C genes. One isolate, *S*. *enterica* ser. Virchow, carrying the *bla*_CTX-M-15_ gene from the breeder farm was ceftiofur resistant. Pulsed-field gel electrophoresis (PFGE) showed that 52.0% *S*. *enterica* ser. Montevideo and 29.6% *S*. *enterica* ser. Albany isolates sourced from the downstream of hatcheries along the broiler chicken supply chain carried the same PFGE types as those of the hatcheries. Thus, the hatcheries showed a high prevalence of *Salmonella* isolates with high antimicrobial resistance and no susceptible isolate. The AMR isolates from hatcheries originating from breeder farms could disseminate to the final retail market along the broiler chicken supply chain. The emergence of AMR *Salmonella* in hatcheries may be due to the horizontal spread of resistant isolates. Therefore, *Salmonella* control in hatcheries, particularly its horizontal transmission, is important.

## 1. Introduction

Salmonellosis, caused by *Salmonella enterica* (*S. enterica*), is among the most frequently reported foodborne bacterial diseases [1]. Contaminated poultry and its products are a major source of motile Salmonellae causing salmonellosis in human worldwide [2,3]. In South Korea, *Salmonella* is the leading (23%) cause of bacterial foodborne poisoning. The annual production of chicken meat, the second-largest source of animal protein, was 957,000 metric tons in 2019, indicating an increase of about 1.6% since 2018 [4,5]. Among them, broilers represented 77% of slaughtered chickens in 2018. Contamination of poultry may occur throughout the broiler production chain, and the potential risk for contamination at each stage has been identified [6].

Humans are likely to be exposed to antimicrobial-resistant (AMR) *Salmonella*, which results from the use of antimicrobials in animals, through contaminated food, thus, leading in a health threat [7]. In recent years, increases in the emergence and spread of AMR *Salmonella*, particularly multidrug-resistant (MDR) *Salmonella*, in humans and animals have been reported worldwide, making it a global challenge [8,9,10]. Therefore, fluoroquinolones (FQs) and third generation cephalosporins (3GC) have become critically important for treating salmonellosis in humans [11]. Thus, *Salmonella* resistant to FQs and 3GC frequently arise in animals with easy dissemination across the food chain [12]. The dissemination of AMR *Salmonella* through the food chain, particularly through chickens, has important implications for the failure of salmonellosis treatment, thus, creating an increased risk to public health by the spread of AMR *Salmonella* via chickens [13].

It is essential to inhibit microorganisms in the broiler chicken supply chain to produce hygienic chicken meat. Epidemiological studies have clearly shown the transmission pathway of *S. enterica* and its serovars associated with poultry and the challenges posed by their virulence and antimicrobial resistance profiles [14]. Consequently, this pathogen has become one of the main targets for the implementation of control strategies along the poultry production chain. Breeding flocks, hatcheries, rearing farms, and slaughter plants are all recognized as critical focal points for managing the risk associated with salmonellae. A hatchery plays an important role in collecting hatching eggs from the upper breeder farm and selling newly hatched chicks to a commercial broiler farm. Some *Salmonella* serovars can persist in hatcheries longer than others, probably due to their ability to form biofilms [15]. Hazard Analysis and Critical Control Point (HACCP) has been applied to poultry farms (including broiler and breeder farms) and chicken slaughterhouses in Korea. However, it has not been applied to hatcheries, thus, warranting a systematic investigation and evaluation of hatchery hygiene, which has not yet been performed [16]. 

*Salmonella* can be introduced into hatcheries by horizontal and vertical transmission routes. The newly hatched chicks are more susceptible to *Salmonella* infection than older birds; as their intestinal flora and immune system are immature, they may become infected with a challenge of 10–100 *Salmonella* cells [17]. A high prevalence of *Salmonella* in one-day-old chickens from hatcheries has been previously reported [18]. *Salmonella* contamination in a hatchery can produce poor-quality chicks, resulting in a decreased feed conversion rate, increased mortality, and poor flock uniformity [16]. Moreover, the prevalence of *Salmonella* in hatcheries is related to *Salmonella* prevalence in derived meat products during processing [19]. Prevention of *Salmonella* contamination in chicken products requires detailed knowledge of the major sources of contamination. The critical role of a hatchery in disseminating *Salmonella* to commercial broiler farms and possibly exposing breeder flocks to contamination on egg trays, trolleys, and vehicles has also been previously reported [20,21,22]. Most of these works have focused on the potential for cross-contamination and infection caused by *Salmonella* in chicks during incubation. Considering that most integrated companies show vertical integration in Korea with numerous potential sources of *Salmonella* contaminants in this system, *Salmonella* control in integrated broiler chicken operations is complicated [23]. 

It is necessary to investigate the occurrence and antimicrobial resistance of *Salmonella* in the poultry production chain as it may aid the optimization of HACCP strategies and reduce the incidence of salmonellosis in humans. In recent years, several reports describing *Salmonella* prevalence in the integrated broiler supply chain in Korea have been published [19,24,25,26]. However, studies focusing on the dissemination or tracing of the AMR *Salmonella* along an integrated broiler chicken operation are limited. The dissemination of AMR *Salmonella* in the broiler farm, slaughterhouse, and its downstream retail markets has been previously described, possibly contributing to the original dissemination of AMR *Salmonella* to retail chicken meat [6,27]. The high prevalence of AMR *Salmonella* colonized in newly hatched chicks in broiler farms has emphasized upstream breeder farm and hatchery as the sources of AMR *Salmonella* in broiler chickens. The purposes of the present study were to identify AMR *Salmonella* occurrence in hatcheries and their upstream breeder farms and reveal the dissemination in a vertically integrated broiler chicken operation in South Korea.

## 2. Materials and Methods 

### 2.1. Sample Collection

There were no vulnerable populations involved, and no endangered species were used in the experiments. No chickens were killed; cloacal swab samples were taken by a veterinarian with prior consent of the farm managers. The present study did not require ethical approval.

From September 2015 to August 2016, 420 fresh samples were collected from 44 hatcheries and their upstream 25 breeder farms. The sampling was done in two parts as follows: (i)A total of 220 cloacal swab test samples were collected from 44 hatcheries. Twenty-five cloacal swab samples were randomly collected from the entire area of each hatchery, and samples from five chicks were pooled into one test sample (S, *n* = 5).(ii)A total of 200 test samples, including 125 cloacal swab samples and 75 litter samples, were collected from 25 breeder farms. Twenty-five cloacal swab samples and fifteen litter samples were randomly collected from the entire areas of each breeder farm, and five samples obtained from the similar area were pooled into one test sample. Finally, cloacal swabs (S, *n* = 5) and litter (L, *n* = 3) were collected from each farm.

### 2.2. Isolation and Identification of Salmonella

Samples were collected in sterile plastic conical tubes (50 mL; SPL Life Sciences Co., Ltd., Seoul, Korea) and stored under refrigeration in the laboratory until analysis, which was performed within 48 h of arrival. *Salmonellae* were isolated using previously described standard methods [28]. The DNA of *Salmonella*-positive colonies, extracted by the boiling method, was further tested by polymerase chain reaction (PCR) using the *Salmonella*-specific gene (*inv*A) [29]. All strains were serotyped per the Kauffmann–White scheme using slide agglutination with O and H antigen-specific sera (BD Difco, Sparks, MD, USA and Denka Seiken Co., Ltd., Tokyo, Japan) [30].

### 2.3. Antimicrobial Susceptibility Test

The minimum inhibitory concentrations (MICs) of the 16 antimicrobials were determined using the KRNV4F Sensititre panel (TREK Diagnostic Systems, Incheon, Korea). The MIC of enrofloxacin was determined using the agar dilution method. *Escherichia coli* (ATCC 25922) was used as the quality control strain. The susceptibility breakpoints of most antimicrobials were interpreted based on the Clinical and Laboratory Standard Institute (CLSI) guidelines [31], whereas those of the antimicrobials used for the animals, including ceftiofur, enrofloxacin, and florfenicol, were interpreted based on the CLSI standards document VET-01 [32]. The CLSI criteria were not available for streptomycin or neomycin, for which we used other references [7,33] (see Appendix A, Appendix A). *Salmonella* isolates, resistant to at least three antimicrobial classes, were considered to be MDR.

### 2.4. Molecular Characterization of Resistance

For determining the molecular characterization of quinolone resistance genes, the quinolone-resistant isolates were further detected on plasmid-mediated quinolone resistance (PMQR) genes by PCR and sequencing. These genes included *qnr*A, *qnr*B, *qnr*S, *qnr*D, *qep*A, *oqx*A, and *aac*(6′)-*lb*-*cr* and mutations in the quinolone-resistance determining region (QRDR), *gyr*A, and *par*C genes, as previously described [34]. Positive controls were used in all PCR reactions. PCR products were purified using the QIAquick Gel Extraction kit (Qiagen, Hilden, Germany) and were further sequenced directly (SolGent Co., Ltd., Daejeon, Korea) for sequence analysis and aligned using the BLAST program (www.ncbi.nlm.nih.gov/BLAST/). The inferred amino acid sequences of the QRDR-encoding genes were compared with the corresponding regions of the reference strain *S.* Typhimurium LT2 (GenBank accession no. AE006468).

For detecting the molecular characterization of resistance to 3GC, isolates exhibiting extended-spectrum β-lactamase/ampicillin-class C (ESBL/AmpC) phenotypes were further screened by PCR, as previously described [35]. 

### 2.5. Pulsed-Field Gel Electrophoresis and BioNumerics Analysis

The *Salmonella* isolates (*n* = 38) selected from the hatcheries and upstream breeder farms were genotyped using pulsed-field gel electrophoresis (PFGE) following the protocols of the Centers for Disease Control and Prevention available on PulseNet (www.pulsenetinternational.org) with certain modifications, as described previously [28]. Among them, *S*. *enterica* ser. *Albany* (*n* = 17), *S*. *enterica* ser. Montevideo (*n* = 9), and *S*. *enterica* ser. Virchow (*n* = 2) were selected to further compare the genotypic relatedness of isolates with relevant serovars (*S*. *enterica* ser. Montevideo, *n* = 100; *S*. *enterica* ser. Albany, *n* = 71; and *S*. *enterica* ser. Virchow, *n* = 25) from downstream of the integrated broiler chicken operation, including broiler farms, slaughterhouse, and retail markets (in our previous studies) [6,27]. 

### 2.6. Statistical Analysis

The Chi-square test was used to test for significant differences in the rates of *Salmonella* isolation among hatcheries and breeder farms and *p*-values less than 0.05 were considered to be statistically significant. The software SPSS (version 19.0; IBM Co., Armonk, NY, USA) was used for statistical analysis.

## 3. Results 

### 3.1. Prevalence and Serovars of Salmonella

A total of 36 isolates (16.4%) from the samples obtained from hatcheries and six isolates (3.0%) obtained from their upstream breeder farms were positive for *Salmonella* (Appendix A). There was a significant difference (*p* < 0.05) in isolation rates among the breeder farm and hatchery; however, there were no significant differences in isolation rates among the cloacal swabs samples (1/125, 0.8%) and the litter samples (5/75, 6.7%) in the breeder farms. Three of 25 (12.0%) breeder farms were positive for *S. enterica*; eighteen of 44 hatcheries (40.9%) were positive for *Salmonella*. The prevalence of *S. enterica* per hatchery and in breeder farms is shown in Appendix A. Five serovars were identified among the 42 *Salmonella*-positive isolates (Table 1). The most common serotype recovered from the hatcheries was *S. enterica* ser. Albany (17 isolates, 47.2%), followed by *S. enterica* ser. Montevideo (11 isolates, 30.6%) and *S.*
*enterica* ser. Senftenberg (five isolates, 13.9%). Only one isolate with *S*. *enterica* ser. Omuna was detected, and two isolates (5.6%) marked as “untypable” could not be assigned to specific serotypes. The serotypes recovered from breeder farms were *S. enterica* ser. Montevideo (3 isolates, 50%) and *S. enterica* ser. Virchow (three isolates, 50%). 

### 3.2. Antimicrobial Susceptibility Analysis

All 36 isolates from hatcheries were resistant to at least one antimicrobial; 19 of these isolates (52.8%) were MDR (Table 2). Among these 19 MDR isolates were all isolates of *S. enterica* ser. Albany (*n* = 17) and the only isolate of *S*. *enterica* ser. Omuna (*n* = 1) and untypable (*n* = 1); no isolate of *S. enterica* ser. Senftenberg or *S. enterica* ser. Montevideo was MDR. Isolates of *S. enterica* ser. Albany with resistance to NAL (17/17, 100.0%), TET (17/17, 100.0%), AMP (17/17, 100.0%), SXT (17/17, 100.0%), and CHL (17/17, 100.0%) were the most prevalent, followed by those resistant to STR (11/17, 64.7%) and FFN (3/17, 17.6%). Isolates of *S. enterica* ser. Montevideo with resistance to NAL (11/11, 100.0%) were the most prevalent, followed by the one resistant to SXT (1/11, 9.1%). Isolates of *S. enterica* ser. Senftenberg with resistance to NAL (5/5, 100.0%) were the most prevalent. Among all isolates from hatcheries, 50.0% (18/36) and 47.2% (17/36) showed intermediate resistance to CIP and enrofloxacin (ENR), respectively. All 36 isolates from the hatcheries were susceptible to the four antimicrobials NEO, GEN, CEP, and XNL. Six antimicrobial resistance profiles were observed among *Salmonella* isolates from the hatcheries; the antimicrobial resistance profile NAL (13/36, 36.1%) was the most prevalent antimicrobial resistance profile, followed by NAL-STR-TET-SXT-AMP-CHL (12/36, 33.3%) (Table 3). The isolates of *S. enterica* ser. Albany were all resistant to ≥5 antimicrobials. Antimicrobial resistance profiles of NAL-STR-TET-SXT-AMP-CHL (*n* = 11), NAL-TET-SXT-AMP-CHL (*n* = 3), and NAL-TET-SXT-AMP-CHL-FFN (*n* = 3) were confirmed. Five of six isolates from breeder farms were MDR.

### 3.3. Quinolone-Resistance Determining Region (QRDR) Mutations, Plasmid-Mediated QuinoloneRresistance (PMQR), and Extended-Spectrum β-Lactamase (ESBL)-Producing Isolates

The prevalence of PMQR and QRDR mutations among *Salmonella* isolates (ENR, MIC ≥ 0.25) is shown in Table 4. PMQR genes were not observed in any test isolate; all test isolates from hatcheries (*n* = 33) showed single point mutations in *gyr*A and *par*C genes. Point mutations with Ser-83-Tyr, Ser-83-Phe, and Asp-87-Gly were found in the *gyr*A gene; point mutations with Tyr-57-Ser and Tyr-57-Thy were found in the *par*C gene. Of the two *Salmonella* isolates from breeder farm with resistance to XNL (MICs ≥ 8), one isolate carried an ESBL gene, which was *bla*_CTX-M-15_.

### 3.4. Correlations among Salmonella Isolates from Hatcheries and Downstream Stages along an Integrated Broiler Chicken Operation Based on Genotypic Characteristics

A total of 38 selected isolates were subdivided into 17 PFGE types with 100% similarity (Figure 1). Different serotypes were detected on the same farms. *S*. *enterica* ser. Montevideo and *S*. *enterica* ser. Virchow isolates were identified from farm BY, *S*. *enterica* ser. Albany and *S*. *enterica* ser. Montevideo from farm SJ, and *S*. *enterica* ser. Omuna and *S*. *enterica* ser. Albany isolates from farm JZ. Isolates, including 17 *S*. *enterica* ser. Albany and 13 *S*. *enterica* ser. Montevideo, were classified into five and six PFGE types, respectively, based on characterization with PFGE using the restriction enzyme *Xba*I. Among these, one PFGE type (M1) was shared among breeder farms and hatchery isolates. Isolates of the same PFGE type were observed in different hatchery farms (type M3 in farms HS and HNL; type M5 in farms HNL and JHG; type M6 in farms YW, HSH, SJ, and ZE; type S in farms NSZY and XH; type A1 in farms YWG and HS; type A2 in farms YY, SJ, and ZH; type A3 in farms HNL and NS; type A4 in farms YWG, HS, and HNL; and type A5 in farms ZH, YJ, NS, and JZ). Conversely, isolates of different PFGE types were observed in the same farms (types M3, M5, A3, and A4 in farm HNL; types A1 and A4 in farms YWG and HS; and types A2 and A5 in farm ZH). 

The 224 isolates of *S. enterica* (*S*. *enterica* ser. Montevideo, *n* = 109; *S*. *enterica* ser. Albany, *n* = 88; and *S*. *enterica* ser. Virchow, *n* = 27) from the integrated broiler chicken operation can be divided into 50 PFGE types, including 30, 11, and 9 PFGE types in *S*. *enterica* ser. Montevideo, *S*. *enterica* ser. Albany, and *S*. *enterica* ser. Virchow isolates, respectively (Appendix A). The distribution of PFGE types of *Salmonella* isolates from different stages along the chicken production chain is shown in Table 5. Eleven PFGE types of *Salmonella* isolates from hatcheries were detected. Among these, two (18.2%) PFGE types (SM-3 and SM-7) were found consistent with upstream breeder farms and eight (72.7%) PFGE types (SM-3, SM-7, SM-11, SM-12, and SM-13; SA-9, SA-10, and SA-11) were consistent with downstream stages, including broiler farms (SM-3, SM-7, SM-11, SM-12, and SM-13), slaughterhouses (SM-7, SM-12, SA-10, and SA-11), and retail markets (SM-7, SM-12, SA-9, SA-10, and SA-11). Among these PFGE types, SM-7 was found in isolates from breeder farms, hatcheries, broiler farms, slaughterhouses, and retail markets (Appendix A). Furthermore, the types SA-10 and SA-11 were also found in isolates from hatcheries, slaughterhouses, and retail markets (Appendix A); no identical PFGE type was found in isolates of *S. enterica* ser. Virchow from breeder farms and the downstream stages (Appendix A). The 52 (52.0%, 52/100) *S*. *enterica* ser. Montevideo isolates procured from the downstream of the hatchery stage carried the same PFGE types as those of the hatchery, and the most prevalent PFGE types were SM-7 and SM-12, accounting for 25.0% (25/100) and 14.0% (14/100) of the isolates, respectively. Similarly, the 21 (29.6%, 21/71) *S*. *enterica* ser. Albany isolates procured from the downstream of the hatchery stage carried the same PFGE types as those of the hatchery, and the second most prevalent PFGE type SA-11 accounted for 19.7% (14/71) of the isolates (Table 6 and Appendix A). The prevalence of overlapped PFGE types in *S. enterica* ser. Montevideo (52/100, 52.0%) is significantly higher (*p* = 0.003) than that of in *S. enterica* ser. Albany (21/71, 29.6%); the prevalence of the most prevalent overlapped PFGE types in each serotype (SM-7 of *S. enterica* ser. Montevideo and SA-11 of *S. enterica* ser. Albany, respectively) is significantly higher than that of non-overlapped PFGE types (SM-6 (or SM-15, 22, 26, 28) and SA-4, respectively) (Appendix A).

## 4. Discussions 

In the present study, the prevalence of *S. enterica* (16.4%) was higher in hatcheries than in its upstream breeder farm (3.0%), despite fumigation being routinely used during hatching (Appendix A) [16]. The prevalence of *Salmonella* in hatcheries varied widely from operation to operation (6.78–44.9%) and may have been associated with differences in hygiene and sanitation levels of each operation and the different detection methods used in each study [25,36,37]. Herein, the *Salmonella* isolation rate from breeder farms was relatively lower (3.0%) than in previous studies in Korea (14.7–19.0%) and China (10.53–18.15%); however, even infected breeder flocks have been shown to cause widespread *Salmonella* contamination [18,25]. The comparison of our *Salmonella* isolation rates in cloacal swabs (1/125, 0.8%) and litter (5/75, 6.7%) samples from breeder farms with previous studies (0% and 40%, respectively) indicates that different sample types may also be a factor influencing the prevalence of *Salmonella* [24,38]. 

In total, *S*. *enterica* ser. Albany was the dominant serovar in hatcheries (Table 1). These results are consistent with our previous study on isolating AMR *Salmonella* isolates from chicken slaughterhouses and retail markets [6]. Similarly, studies conducted on the prevalence of *Salmonella* in poultries in Vietnam, Malaysia, and Myanmar showed that the predominant serovar was *S*. *enterica* ser. Albany (34.1%, 35.4%, and 38%, respectively) [39,40,41]. Contrary to our results, *S*. *enterica* ser. Hadar was the most frequently reported serovar in integrated broiler operations in Korea; the *Salmonella* serotype most often isolated from the hatchery was *S*. *enterica* ser. Senftenberg [16,19,25]. *S*. *enterica* ser. Albany, one of the prominent serovars in poultry that has been infecting animals and humans for several decades, may be an emerging serotype in Korea in the future [42,43]. Consistent with our results, *S*. *enterica* ser. Montevideo, *S*. *enterica* ser. Senftenberg, and *S. enterica* ser. Virchow were the most frequently reported serovars in the poultry industry in Korea [25,26,44]. *S*. Enteritidis has typically been the most common serotype responsible for *Salmonella* infections in poultry in Korea for several decades; however, this prevalence has decreased, and this serovar has been gradually replaced by other emerging serovars. Therefore, the predominant *Salmonella* serovar varies from company to company and time to time.

Antimicrobial resistance of *Salmonella* is a globally emerging problem of public health concern. In the present study, no susceptible isolate was found in any of the 42 isolates. Among the 16 antimicrobial agents tested, the highest resistance rate observed in the hatchery was to NAL (97.2%), followed by SXT (55.6%), AMP (52.8%), TET (50.0%), and CHL (50.0%), which is consistent with previous reports (Table 2) [6,45]. Antimicrobial resistance was detected even in isolates from hatcheries that were not treated with antimicrobials. One potential explanation is that those AMR isolates came from upstream breeder farms. Quinolones, ampicillin, and tetracyclines have been widely used for therapy, prophylaxis, and growth promotion by breeders, while sulfonamides have been used in human and veterinary medicine for 40 years [46,47]. Another potential explanation is that the hatcheries are contaminated with AMR *Salmonella* in the internal environment of the hatchery or the external natural environment [48]. Contrary to the increasing incidence of FQ-resistant *Salmonella* reported worldwide, we did not find FQ resistance in the present study. However, a marked resistance to NAL reported herein could be a matter of concern because NAL resistance has been associated with a decrease in susceptibility to FQs, which are used to treat salmonellosis in humans. The present study indicated that MDR *Salmonella* contamination was more widespread in the hatchery (19/36, 52.8%) (Table 2) than suggested by another report [49]. Among MDR isolates, we observed resistance to 3GC and COL, which are critically important in treating salmonellosis in humans [11]. For example, two *S*. *enterica* ser. Virchow isolates from breeder farms were resistant to XNL, with the antimicrobial resistance profiles NAL-NEO-STR-TET-AMP-XNL and NAL-NEO-TET-AMP-XNL (Table 3); one isolate (untypable) from the hatchery was resistant to COL, with the antimicrobial resistance profile SXT-AMP-FOX-COL. Colistin, as an antimicrobial substance, was used against Gram-negative bacteria. The use of colistin has been limited due to systemic toxicity. However, it has been re-introduced as a last-line option in the treatment of human infections [50]. The resistance can be transmitted to humans through the food chain. Eventually, it can lead to microbial cross-resistance and pose a threat to human health. Therefore, it is mandatory to monitor the dissemination of resistance to colistin [51].

Although PMQR has been studied and increasingly reported, QRDR mutations seem to represent the main mechanism of quinolone resistance in animal isolates [52]. Moreover, PMQR was commonly detected in *Enterobacteriaceae*, particularly in *E. coli*, and the prevalence of PMQRs in *Salmonella* remains extremely low [53]. This finding was consistent with our observations that the global level of FQ non-susceptibility may be mainly due to QRDR chromosomal mutations (Table 4). Our results indicated that missense mutations frequently occurred in the QRDR of *gyr*A and *par*C, which are considered to be the major quinolone resistance determinants in *Salmonella* [54]. In the present study, we identified QRDR point mutations in the *gyr*A and *par*C genes in all selected *Salmonella* isolates from hatcheries, a finding inconsistent with previous studies showing one-point mutation only in the *gyr*A gene as the main pattern [55,56,57]. The results imply that resistance to FQs is continuously evolving with time. The results also show that antimicrobial pressure, rather than the horizontal transmission of antimicrobial resistance genes in chickens, leads to the appearance of antimicrobial resistance; clonal dissemination seems to be a key contributing factor for increasing resistance to FQs among *Salmonella* in hatcheries, where antimicrobials are not applicable [58]. This finding indicates the potential risk that *Salmonella* isolates with mutations in *gyr*A and *par*C could naturally be maintained during hatching, even with no antimicrobial pressure. The *Salmonella* isolates with QRDR may be directly transmitted to the downstream broiler farms through their carrying by day-old chicks [59]. Our results also showed that one resistance gene (*bla*_CTX-M-15_), which encodes resistance to ESBL, was identified in *S*. *enterica* ser. Virchow, one of the most frequently identified serotypes in 2015–2016 [6]. There was a potential risk of ESBL-producing *Salmonella* isolates being transmitted to humans through contaminated poultry products [60]. 

Currently, PFGE is an easy and effective method to assess relatedness among *Salmonella* isolates from different sources [6,27]. The clonal relationship among isolates from hatcheries and their upstream breeder farms at the chromosome level was accessed using PFGE (Figure 1). There was frequent *Salmonella* cross-contamination among hatcheries and among hatcheries and upstream breeder farms. An identical PFGE type (M1) was shared between isolates from a hatchery and its upstream breeder farm isolates, suggesting that *Salmonella* contamination in hatcheries could be achieved by a direct vertical top-down transmission [36]. Isolates from different hatcheries shared the same PFGE types, indicating *Salmonella* cross-transmission among the hatcheries, possibly due to the sharing of the same source of eggs or trucks within the same operation [24]. 

To further determine and compare the genotypic relatedness of isolates from the integrated broiler chicken operation, selected isolates of the three most prevalent serotypes from breeder farms, hatcheries, broiler farms, slaughterhouses, and retail markets were analyzed by PFGE (Appendix A). A highly consistent PFGE pattern (SM-7) from different sources revealed that the AMR *S. enterica* clones could disseminate through the broiler chicken supply chain (Table 6 and Appendix A). The SM-7, not the main PFGE type in hatcheries, could be disseminated to the downstream stage (broiler farm) even throughout the broiler supply chain, suggesting that *S*. *enterica* ser. Montevideo could persist and is difficult to eliminate from the environment, probably due to its biofilm-producing ability [61,62]. Herein, 52% of the *S*. *enterica* ser. Montevideo and 29.6% of the *S*. *enterica* ser. Albany isolates from the downstream of the hatchery carried the same PFGE types as those of the hatchery, indicating that *Salmonella* contamination in hatcheries was an important source of *Salmonella* contamination in the integrated broiler chicken operation (Table 6 and Appendix A). From Appendix A, it can be concluded that biocontrol of *Salmonella* contamination in hatcheries is important; and control of *S. enterica* ser. Montevideo isolates from hatcheries are more necessary.

The routes of *Salmonella* cross-contamination within the hatchery and between the hatchery and its upstream breeder farm were complicated. *Salmonella* was detected in the hatchery and its upstream breeder farm (represented by *S*. *enterica* ser. Montevideo) (Table 5). Two PFGE types (SM-3 and SM-7) were consistent with upstream breeder farms, indicating that vertical transfer of infection from breeding birds to progeny is an important aspect of the epidemiology of *S. enterica* infection within the poultry industry [24,63]. More importantly, emerging serotypes and genotypes were detected in the hatchery. The hatchery samples were contaminated with non-original *Salmonella* with a serotype different from those in the breeder farm (represented by *S*. *enterica* ser. Albany); the hatchery samples were contaminated with non-original *Salmonella* with PFGE types different from those in the breeder farm (represented by SM-11, SM-12, SM-13, and SM-14). *Salmonella* was absent in upstream breeder farms but present in the hatchery, suggesting that contamination can happen during hatching. Therefore, apart from the original *Salmonella* contamination in upstream breeder farms, there is at least one more route of *Salmonella* contamination in the hatchery. There are many ways in which *Salmonella* can enter these extensive and integrated operations and be recirculated and amplified by various routes [26]. In some cases, clonal horizontal transmission in the hatchery and on the farm during the rearing period is of greater importance and leads to the isolation of a greater variety of *Salmonella* serovars [64]. For instance, several risk factors for horizontal transmission have been identified, such as inadequate cleaning and disinfection of hatching houses, which leads to contamination of the downstream hatching eggs and a poor level of hygiene [64]. Usually, *Salmonella* infection does not interfere with hatchability, but during hatching, the organisms are widely spread in the hatcher via ducts and the fluff that is disseminated by forced ventilation [63,65]. The higher prevalence of *Salmonella* in a hatchery than in a relatively clean place indicates that intervention strategies must target this stage to prevent *Salmonella* from entering the downstream broiler farm. This approach requires that *Salmonella* should be detected quickly and accurately at the hatching stage before entering the broiler farm. Our study had the limitation that we did not provide direct evidence to prove the horizontal transmission of *Salmonella* in the hatchery. Further studies focusing on the investigation of the horizontal transmission routes of *Salmonella* in hatcheries are needed.

## 5. Conclusions 

Our study found a high prevalence of *Salmonella* at the hatchery stage, high antimicrobial resistance, and no susceptible isolates. The AMR isolates could disseminate to the downstream stage, even to the final retail market, along the broiler chicken supply chain. Compared with breeder farms, *Salmonellae* in the hatchery play a more important role in their emergence and spread in the broiler production chain. The emergence of AMR *Salmonella* in the hatchery may be due to the horizontal spread of resistant isolates rather than the first contamination of *Salmonella,* and then the acquisition of resistance. Therefore, the presence of AMR *Salmonella* on day-old chicks in the hatchery is very likely to reduce the effectiveness of the biosafety prevention and control of *Salmonella* in the rearing farm. The broiler farm, with a high possibility to explore antimicrobials, could provide new or higher resistant isolates in the next stage along the broiler production chain. Using *Salmonella*-free chicks would be an effective way to control AMR *Salmonella* emergence and transmission in broiler chickens. Furthermore, the national surveillance program should be implemented in hatcheries for monitoring AMR *Salmonella* in chickens. 

## Figures and Tables

**Figure 1 animals-11-00154-f001:**
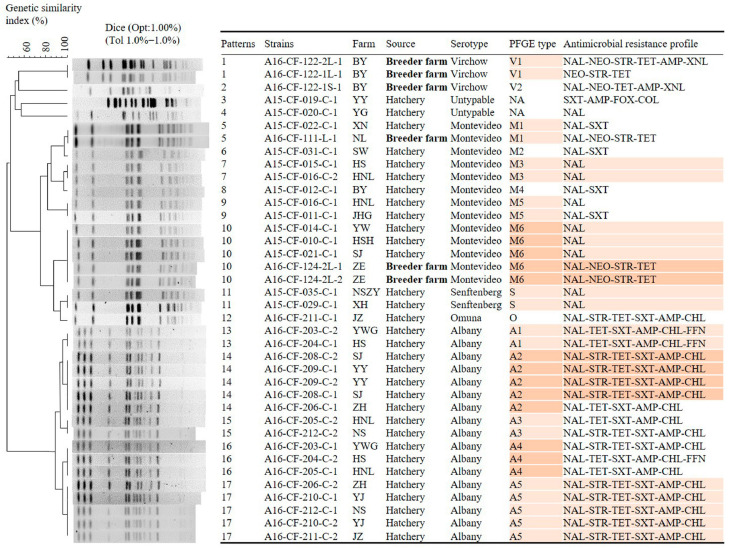
Dendrograms of the pulsed-field gel electrophoresis (PFGE) types of the 38 selected *Salmonella* isolates recovered from the hatchery and upstream breeder farms. These isolates and their association with source and antimicrobial resistance profiles are shown. Backgrounds with the same-colored bars (orange) indicate samples with the same PFGE patterns and antimicrobial resistance profiles; bold and non-bold indicate samples with different source.

**Table 1 animals-11-00154-t001:** Serotype distribution of *Salmonella enterica* in hatcheries and their upstream breeder farms.

Serovar (Serogroup)	Sampling Site, *n* ^a^ (%)		
	Hatchery	Breeder Farm	Total
Albany (C2–C3)	17 (47.2)	0	17 (40.5)
Montevideo (C1)	11 (30.6)	3 (50.0)	14 (33.3)
Senftenberg (E4)	5 (13.9)	0	5 (11.9)
Virchow (C1)	0	3 (50.0)	3 (7.1)
Omuna (C1)	1 (2.8)	0	1 (2.4)
Untypable	2 (5.6)	0	2 (4.8)
Total	36	6	42

^a^*n*, number of isolates.

**Table 2 animals-11-00154-t002:** Antimicrobial resistance of *Salmonella enterica* isolated from hatcheries and their upstream breeder farms ^a^.

Serovar	*n*	No. (%) of Isolates Resistant to Antimicrobials																
		NAL	CIP	ENR	NEO	GEN	STR	TET	AMC	CEP	FOX	XNL	AMP	SXT	COL	FFN	CHL	MDR
Hatchery																		
Albany	17	17 (100.0)	0	0	0	0	11 (64.7)	17 (100.0)	0	0	0	0	17 (100.0)	17 (100.0)	0	3 (17.6)	17 (100.0)	17 (100.0)
Montevideo	11	11 (100.0)	0	0	0	0	0	0	0	0	0	0	0	1 (9.1)	0	0	0	0
Senftenberg	5	5 (100.0)	0	0	0	0	0	0	0	0	0	0	0	0	0	0	0	0
Omuna	1	1 (100.0)	0	0	0	0	1 (100.0)	1 (100.0)	0	0	0	0	1 (100.0)	1 (100.0)	0	0	1 (100.0)	1 (100.0)
Untypable	2	1 (50.0)	0	0	0	0	0	0	0	0	1 (50.0)	0	1 (50.0)	1 (50.0)	1 (50.0)	0	0	1 (50.0)
Subtotal	36	35 (97.2)	0	0	0	0	15 (41.7)	18 (50.0)	0	0	1 (2.8)	0	19 (52.8)	20 (55.6)	1 (2.8)	3 (8.3)	18 (50.0)	19 (52.8)
Breeder farm																		
Montevideo	3	3 (100.0)	0	0	3 (100.0)	0	3 (100.0)	3 (100.0)	0	0	0	0	0	0	0	0	0	3 (100.0)
Virchow	3	2 (66.7)	0	0	3 (100.0)	0	2 (66.7)	3 (100.0)	0	0	0	2 (66.7)	2 (66.7)	0	0	0	0	2 (66.7)
Subtotal	6	5 (83.3)	0	0	6 (100.0)	0	5 (83.3)	6 (100.0)	0	0	0	2 (33.3)	2 (33.3)	0	0	0	0	5 (83.3)
Total	42	40 (95.2)	0	0	6 (14.3)	0	17 (40.5)	24 (57.1)	0	0	1 (2.4)	2 (4.8)	21 (50.0)	23 (54.8)	1 (2.4)	3 (7.1)	18 (42.9)	24 (57.1)

^a^ NAL, nalidixic acid; CIP, ciprofloxacin; ENR, enrofloxacin; NEO, neomycin; GEN, gentamicin; STR, streptomycin; TET, tetracycline; AMC, amoxicillin/clavulanic acid; CEP, cephalexin; FOX, cefoxitin; XNL, ceftiofur; AMP, ampicillin; SXT, trimethoprim/sulfamethoxazole; COL, colistin; FFN, florfenicol; CHL, chloramphenicol; MDR, multiple drug resistance; *n*, number of isolates.

**Table 3 animals-11-00154-t003:** Antimicrobial resistance profile of *Salmonella* isolates from hatcheries and their upstream breeder farm.

No.	Antimicrobial Resistance Profile ^a^	Hatchery (*n* ^b^ = 36)		Breeder Farm (*n* = 6)	
		*n* (%)	Serovars (*n*)	*n* (%)	Serovars (*n*)
1	NAL	13 (36.1)	Montevideo (7), Senftenberg (5), untypable (1)	0	-
2	NAL-STR-TET-SXT-AMP-CHL	12 (33.3)	Albany (11), Omuna (1)	0	-
3	NAL-SXT	4 (11.1)	Montevideo (4)	0	-
4	NAL-TET-SXT-AMP-CHL	3 (8.3)	Albany (3)	0	-
5	NAL-TET-SXT-AMP-CHL-FFN	3 (8.3)	Albany (3)	0	-
6	SXT-AMP-FOX-COL	1 (2.7)	Untypable (1)	0	-
7	NAL-NEO-STR-TET	0	-	3 (50.0)	Montevideo (3)
8	NAL-NEO-STR-TET-AMP-XNL	0	-	1 (16.7)	Virchow (1)
9	NEO-STR-TET	0	-	1 (16.7)	Virchow (1)
10	NAL-NEO-TET-AMP-XNL	0	-	1 (16.7)	Virchow (1)

NAL, nalidixic acid; NEO, neomycin; STR, streptomycin; TET, tetracycline; FOX, cefoxitin; XNL, ceftiofur; AMP, ampicillin; SXT, trimethoprim/sulfamethoxazole; COL, colistin; FFN, florfenicol; CHL, chloramphenicol; *n*, number of isolates.

**Table 4 animals-11-00154-t004:** Prevalence of plasmid-mediated quinolone resistance (PMQR) and quinolone-resistance determining region (QRDR) mutations among *Salmonella* (enrofloxacin (ENR), minimum inhibitory concentrations (MIC) ≥ 0.25) isolated from hatcheries and their upstream breeder farms.

Patterns	ENR MIC (µg/mL)	PMQR	QRDR Mutations							
			*gyr*A			*par*C		No. of Isolates		
			Ser-83-Tyr	Ser-83-Phe	Ser-87-Gly	Tyr-57-Ser	Tyr-57-Thy	Hatchery	Breeder Farm	Total
P1	0.25−0.50	□	□	□	■	■	□	15	3	18
P2	0.25−0.50	□	□	■	□	□	■	9	1	10
P3	0.25−0.50	□	□	■	□	■	□	8	0	8
P4	0.25	□	■	□	□	■	□	1	0	1
P5	0.50	□	□	□	□	■	□	0	1	1
*n*, %		0	1 (2.6)	18 (47.4)	18 (47.4)	37 (97.4)	1 (2.6)	33	5	38

*n*, indicates the number of isolates; % indicates the percentage; ■/□, indicate the presence/absence, respectively, of designated gene and mutation in each *Salmonella* isolate tested.

**Table 5 animals-11-00154-t005:** The distribution of PFGE type of *Salmonella* isolates from different stages along the chicken production chain.

Serovar	PFGE Type	Sampling Site				
		Breeder Farm	Hatchery	Broiler Farm	Slaughterhouse	Retail Market
Montevideo	SM	3 ^b^, 7	3, 7, 11, 12, 13, 14	3, 5, 6, 7, 9, 10, 11, 12, 13, 15, 16, 17, 18, 20, 24, 25, 26, 27, 28, 29, 30	1, 2, 4, 7, 8, 10, 12, 21, 23	7, 12, 19, 22
Albany	SA	- ^a^	6, 8, 9, 10, 11	-	1, 2, 3, 4, 5, 7, 10, 11	1, 2, 4, 5, 9, 10, 11
Virchow	SV	8, 9	-	2	2, 3, 4, 6, 7	1, 2, 4, 5

^a^ No isolate of the target serotype; ^b^ number indicating the PFGE type, as shown in Appendix A.

**Table 6 animals-11-00154-t006:** The prevalence of *Salmonella* isolates from downstream of the hatchery stage along the integrated broiler chicken operation, with overlapped PFGE types with the hatchery stage *.

Serovar	Overlap or Not	PFGE Type	Downstream (Broiler Farm, Slaughterhouse, Retail Market)	
			Number of Isolates/per PFGE Type	Percentage (%)
Montevideo	Overlap	SM-7	25	25.0
		SM-12	14	14.0
		SM-13	5	5.0
		SM-3, 11	4	4.0
		Subtotal	52	52.0
	Non-overlap	SM-6, 15, 22, 26, 28	4	4.0
		SM-27	3	3.0
		SM-1, 5, 10, 16, 19, 23, 29	2	2.0
		SM-2, 4, 8, 9, 17, 18, 20, 21, 24, 25, 30	1	1.0
		Subtotal	48	48.0
		Total	100	100.0
Albany	Overlap	SA-11	14	19.7
		SA-10	4	5.6
		SA-9	3	4.2
		Subtotal	21	29.6
	Non-overlap	SA-4	24	33.8
		SA-7	9	12.7
		SA-1	8	11.3
		SA-5	4	5.6
		SA-2	3	4.2
		SA-3	2	2.8
		Subtotal	50	70.4
		Total	71	100.0
Virchow	Non-overlap	SV-2	11	44.0
		SV-4	6	24.0
		SV-1	4	16.0
		SV-3, 5, 6, 7	1	4.0
		Total	25	100.0

* This table was converted to a Venn diagram to illustrate the distribution of PFGE types in Appendix A.

## Data Availability

The data presented in this study are available on request from the corresponding author.

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
