# Peer review of "The Occurrence of Antimicrobial-Resistant Salmonella enterica in Hatcheries and Dissemination in an Integrated Broiler Chicken Operation in Korea"

_animals, 2021, doi:10.3390/ani11010154_

Round 1

Reviewer 1 Report

General comments:

Ke Shang et al characterized the distribution of AMR in Salmonella population isolated from chicken farms and processing facilities. Their results found high prevalence of AMR in chicken samples. The molecular pattern of AMR indicated hatcheries as an important source of the dissemination of AMR genes. Their findings suggested a key point for bio-control of AMR Salmonella in chicken industry. Overall, the manuscript is well structured and data collection is described in details. However, the results from statistical analysis are neither  described in the text nor indicate in the figures. 

In addition, there are some minor grammar issues and I propose a few changes in the presentation of figures/tables to improve the clarity for readers. See specific comments below.

Specific comments:

Line 28, 29, 32: Species name is not italicized. This is an issue throughout the manuscript. Please check for consistence.

Line 182-188: Are there statistical difference for Salmonella positive rates among different sampling conditions (eg. Source type, locations et al)? The statistic is mentioned in the method, but the statistic results are not provided in the result section.

Similarly, for results describing PFGE pattern, are there statistical difference among sampling sites? These are important information to support authors’ conclusion about the importance of hatcheries in bio-control.

Figure 1. The relationships between PFGE patterns and sample source are not very clear to me. I suggest use a colored bar to indicate samples with the same attribute (eg. Source / Anti-microbial resistance profiles) to direct visualize their distribution.

Table 6. It is a bit difficult for readers to identify the shared PFGE types among different sites. I suggest convert this table to a Venn diagram to illustrate the distribution PFGE types.

Reviewer 2 Report

In Section 2.2, 2.3, and 2.5 (and some of the results), the information presented in the current article is very similar or identical to previous publications from the group. This needs to be addressed.

P1 L30: “Isolates showed no antimicrobial susceptibility.”, consider using “resistance to at least one antibiotic tested".

P1 L31: “Thirty-three isolates (enrofloxacin, MIC ≥ 0.25)”, this information does not indicate the phenotypic pattern of the isolates as MIC ≥ 0.25 can be from sensitive, intermediate resistance to resistant.

P1 L36-38: “The AMR isolates from hatcheries originating from breeder farms”, the data provided does not support this statement.

P3 L102 (and across the article): “in hatcheries and their upstream breeder farms”, consider using the chicken production chain order.

P3 L109: Clarify the association of the 44 hatcheries and their upstream 25 breeder farms.

P3 L111-118: Specify the number of test sample (same for breeder farms). Only one per hatcheries? Were the swabs collected during the same birds’ age? Were the samples from the hatcheries collected at the same time frame as its respective upstream breeder farm?

P4 L143: Please Spell-out the acronym.

P4 L147: Missing MIC value for COL.

P4 L165: Why not 42? Clarify the selection of isolates.

P4 L167: Require reference.

P5 L181: Prevalence and serovars of Salmonella: Those numbers tell that the contamination does not come from the breeder. Also, is that from a multiple hatcheries and breeder farms? Or isolate cases? (Based on Fig 1, 5 isolates from Breeder Farms come from only 2 farms). Consider adding in this section the prevalence rate isolates per hatcheries or breeder farms.

P9 L1: QRDR mutations, PMQR, and ESBL-producing isolates: “(enrofloxacin, MIC ≥ 0.25)”, this information does not indicate the phenotypic pattern of the isolates. Why only 38 tested for QRDR and PMQR? Perhaps include the MIC results, or the phenotypic profile, in the same table?

P9 L16: Please clarify when not all isolates are selected for analysis.

The manuscript requires minor spell check, it is easy to read, and the microbiological and molecular analysis conducted were appropriate. However, the results presented in this manuscript does not support the statement from the authors regarding the origin of AMR isolates from breeder to hatcheries. It is questionable whether the number of isolates from breeder farms and hatcheries are representative of the diversity associated with chickens over a one year period. The number of isolates fingerprinted is very low. Were the serovars found in this study also found in the downstream process? Based on Table 5, it’s clear that it does not come from Breeder farms; and based on the Figure 1, even though the association of breeder farms and hatcheries are not clear in this article, the phenotypes found within breeder farms are not carried on to the hatcheries. It would be beneficial and enhanced the quality of the manuscript if the information presented in references 6 and 27 were combined with the information presented in the current manuscript. For example, it needs to be better summarized/presented whether the AMR profiles were found locations in the poultry production continuum. The authors should have examined the the role of integrons, plasmids, and SXT elements in the multidrug resistance of the isolates found in this study, as well as, explore more the possible mechanisms of introduction of multiple serotypes and phenotypes on a continuing basis, or as a result of a resident population of subtypes within the hatcheries.

Reviewer 3 Report

Revision of manuscript animals-1024603

Dear Authors,

Your manuscript entitled “The occurrence of antimicrobial-resistant Salmonella enterica in hatcheries and dissemination in an integrated broiler chicken operation in Korea” is a very interesting work on distribution of Salmonella among hatchery and broiler production chain. The work is well planned and conducted. Results are clear, well presented and discussed. I particularly appreciate the last part on PFGE characterization of isolates obtained from different level of production chain; it provides important information on Salmonella Epidemiology.

Below only two minor corrections.

  • Matherial and Methods:
    • Lines 148-149: “Salmonella isolates, resistant to at least three antimicrobial agents, 148 were considered as MDR” I disagree with provided definition of MDR because, generally, a strain is considered MDR if it shows resistance in 3 different antimicrobial classes and not to 3 antimicrobial agents (see the follofing article; the same definition of MDR was adopted by EFSA for antimicrobial resistance surveillance program; Magiorakos et al. Multidrug‐resistant, extensively drug‐resistant and pandrug‐resistant bacteria: an international expert proposal for interim standard definitions for acquired resistance. Clin Microbiol Infect . 2012 Mar;18(3):268-81. doi: 10.1111/j.1469-0691.2011.03570.x.).
  • Results
    • Lines 191-207: Please, review the results on MDR considering previous comment; probably nothing will change.
  • .

I sincerely hope that these suggestions will enhance this manuscript. However, if I have made any errors or misinterpretations, I apologize in advance.

Sincerely

The Reviewer
